# COVID-19 Vaccine Coverage and Factors Associated with Vaccine Hesitancy: A Cross-Sectional Survey in the City of Kinshasa, Democratic Republic of Congo

**DOI:** 10.3390/vaccines12020188

**Published:** 2024-02-12

**Authors:** Pierre Z. Akilimali, Landry Egbende, Dynah M. Kayembe, Francis Kabasubabo, Benito Kazenza, Steve Botomba, Nguyen Toan Tran, Désiré K. Mashinda

**Affiliations:** 1Patrick Kayembe Research Center, Kinshasa School of Public Health, University of Kinshasa, Kinshasa P.O. Box 11850, Congo; dirchkayembe@gmail.com (D.M.K.); fkabasu13@gmail.com (F.K.); 2Department of Nutrition, Kinshasa School of Public Health, University of Kinshasa, Kinshasa P.O. Box 11850, Congo; landry.egbende@unikin.ac.cd (L.E.); benito.kazenza@unikin.ac.cd (B.K.); steve.botomba@unikin.ac.cd (S.B.); 3Australian Centre for Public and Population Health Research, Faculty of Health, University of Technology Sydney, P.O. Box 123, Sydney, NSW 2007, Australia; nguyentoan.tran@uts.edu.au; 4Faculty of Medicine, University of Geneva, Rue Michel-Servet 1, 1206 Genève, Switzerland; 5Department of Biostatistics and Epidemiology, Kinshasa School of Public Health, University of Kinshasa, Kinshasa P.O. Box 11850, Congo; desire.mashinda@unikin.ac.cd

**Keywords:** COVID-19, vaccine hesitancy, associated factors

## Abstract

Vaccination against COVID-19 has been the main strategy used by most countries to limit the spread of the virus. However, vaccine uptake has been low in Africa, leading to the implementation of several interventions in order to improve vaccine coverage. This study was conducted due to the lack of information about COVID-19 vaccine coverage and the factors associated with vaccine hesitancy. This cross-sectional study was carried out in Kinshasa city using multi-stage random sampling. A total of 2160 households were included in this study. The data were analyzed using Stata 17 software. The means and standard deviations were computed for continuous data that followed a normal distribution, whereas proportions together with their 95% confidence intervals (CIs) were computed for categorical variables. The connections between dependent variables and each independent variable were tested using either Pearson's chi-square test or Fisher's exact test. The logistic regression method was employed to determine the factors that are linked to hesitation in obtaining the COVID-19 immunization. The majority of respondents were aged between 25 and 34 and 35 and 49 (28.9%). During this study, 15% (95% CI [13.25–17.9]) of respondents had received at least one dose of the COVID-19 vaccine. The prevalence of vaccine hesitancy was 67% (CI95%:64.9–69.1). Among the reasons given for refusing to be vaccinated, most respondents cited concerns about the vaccine being unsafe or causing adverse reactions (45%). Among the reasons given for accepting the vaccine, 26% thought that the vaccine prevented superinfection. The factors associated with hesitancy toward the COVID-19 vaccine were female gender, an age of less than 35 years, and living in non-slum households. Despite the interventions implemented across the country, the reluctance to be vaccinated remains a problem; this could lead to poor health outcomes, especially among the elderly and those with pre-existing conditions. It is important to step up awareness-raising campaigns in the community in order to increase the uptake of vaccination.

## 1. Introduction

Vaccination against COVID-19 has been the main strategy used in the majority of countries to limit the spread of COVID-19 and combat its mortality [1,2,3,4,5]. Studies have been carried out in several developed countries to find effective vaccines that are capable of limiting the spread of the disease [6]. However, acceptance of the vaccine has been low in certain European countries and especially in Africa, leading to the implementation of several interventions in order to improve vaccination coverage [1]. In Africa, the WHO, through the Covax mechanism, has supplied vaccines to a number of countries that were unable to purchase them [7]. This was in order to make vaccines available even to poor populations. It has been difficult for African countries to reach the WHO target, with only 9% of people being fully vaccinated by the end of 2021 [7]. More than 900 million doses of vaccine needed to be administered to bring the continent’s coverage to around 40% [7].

Hesitancy to adopt this strategy has been noted in several countries around the world, including Africa [8,9,10]. The main reasons for this, such as a fear of the effects of the vaccine or an imbalance between the risks and benefits, as well as certain cultural and religious factors, have prevented the vaccine from being accepted [11,12]. In addition, there is a lack of knowledge about both the disease and the vaccine [8,13,14]. Most countries have implemented measures to speed up the administration and acceptance of vaccines; these include making it compulsory to be vaccinated before travelling or taking part in certain competitions, especially sporting events [15]. In some countries, one must be vaccinated before entering public places such as shops, supermarkets, etc., resulting in many people getting vaccinated because other barriers have been lifted [15]. In the Democratic Republic of Congo (DRC), several vaccines have been made available to combat COVID-19. The number of people who are vaccinated against COVID-19 is around 10% of the general population, and around 8% of the population has been fully vaccinated [16,17]. Like in other countries, hesitancy to be vaccinated is also a problem. Ditekemena et al. assessed the willingness of the population to be vaccinated if offered a vaccine and found that 55.9% of participants were willing to be vaccinated against COVID-19 [16]. This could also be linked to the population’s general level of knowledge about COVID-19, as knowledge is a major determinant of adherence to COVID-19 control measures [18].

A number of measures have been implemented throughout the country in order to increase public support for this strategy and combat the spread of the disease [19]. Awareness campaigns were carried out throughout the country to improve access to information and combat all the rumors that were circulating. In addition to these media awareness campaigns, all travelers were asked to be tested on arrival at the airport, except those who were vaccinated [19]. Only fully vaccinated travelers could travel throughout the country without being tested for COVID-19 [19]. However, despite all these interventions, it is difficult to obtain an idea of the factors that lead to low vaccination coverage in order to improve the control strategies that are implemented. Most of the studies carried out have analyzed the intention to be vaccinated and some have focused on the perspective of healthcare workers [20]. These analyses did not include factors that are associated with vaccination coverage and the reasons given for refusing vaccination, as these studies were limited to an assessment of individuals’ intention to be vaccinated. For this reason, this study aimed to assess COVID-19 vaccine coverage and identify the reasons that individuals are hesitant toward these different vaccines within the population.

## 2. Materials and Methods

### 2.1. Settings

According to the Ministry of Health’s 2017 statistical annual that was published in 2019, the city–province of Kinshasa has 35 health zones with 9 general referral hospitals. The city is subdivided into 393 health areas and has 712 first-level structures. However, almost half the population has no access to basic social needs, such as water and hygiene, and there is also malnutrition and food insecurity [18]. According to the National Immunization Program's data as of July 31, 2022, out of the 39,929,390 doses received in the country, 1,445,060 AstraZeneca doses were returned to COVAX. Additionally, 143,300 doses expired at the Central Hub, with 130,000 doses being Turkovac and 13,300 doses being AstraZeneca. Furthermore, 36,825,190 doses were distributed to the provinces. The DRC possessed a total of 6,529,628 doses of a certain substance. This included 1,515,840 doses located in the Central Hub and 5,013,788 doses distributed among the provinces. A total of 2,182,951 doses throughout the provinces had reached their expiration date. The government has administered a total of 19,753,514 doses of the vaccine, as of July 31, with a reported completion rate of 76.3%.

### 2.2. Study Design and Sampling

A cross-sectional study was conducted from 27 July to 3 August 2022 in the city of Kinshasa. The study was carried out in households and the respondents to the questionnaires were either the heads of household or an adult member of the household. All households that were within the selected areas of the city of Kinshasa and had the following characteristics were included in this study.

Sampling adapted from Lot Quality Assurance Sampling (LQAS) was used to select the households. At the first level, the province of Kinshasa was selected on a purposive basis according to the guidelines of the study sponsor and the location of the Kinshasa School of Public Health. At the second level, the health zones were selected using a stratified sampling technique. Based on the list of all health zones, two strata were created; these were made up of urban and rural health zones. Within each stratum, enumeration areas of three health zones were formed; in each, three health zones with the same geographical and socio-demographic characteristics were grouped. Then, in each enumeration area, a single health zone was sampled in a simple random fashion using a random number generator that was provided by Microsoft Excel. In the third stage, in each selected health zone, three health areas were selected in a simple random fashion using the random number generator that is provided by MS Excel. In the fourth stage, the households to be surveyed were selected via systematic sampling after a plot survey was conducted; this enabled households to be enumerated and a sampling frame to be drawn up according to the eligibility criteria. This sampling frame included all the plots in the neighborhood that were selected for the group, which contained at least one eligible statistical unit.

A total of 2160 households were included in this study in order to assess COVID-19 vaccination coverage and the factors associated with non-vaccination.

### 2.3. Measures

The socio-demographic data encompassed variables such as residential area, gender, age, marital status, religion, educational attainment, income level, household size, and housing status in informal settlements (slums) [18]. The categorization of education level was deemed “low” if the individual had not completed secondary education or vocational training, “medium” if they had completed it, and “high” if they had completed higher or university education. The socio-economic status was assessed using a wealth index derived from a range of household assets (such as radio, tape recorder, television set, bicycle, torch, and horse or donkey cart), housing conditions (including roof material, number of rooms, type of wall, windows, availability, and type of latrines), and ownership of domestic animals. The study participants were categorized based on their wealth index score, which was divided into quintiles.

Data on hesitancy toward COVID-19 vaccination included questions relating to individuals’ COVID-19 vaccination status, reasons for being vaccinated, and reasons for not being vaccinated. Vaccine hesitancy was defined as “delay in acceptance or refusal of vaccination despite the availability of vaccination services”. In this study, vaccine hesitancy was defined as the response of “no” or “don’t know/not sure” to whether the participant would get the COVID-19 vaccine as soon as it became available [21].

### 2.4. Data Collection

The data were collected using tablets that were configured with the surveyCTO application. After three days of training on the objectives of the survey and a review of the data collection tools used and ethical aspects of the study, 126 interviewers (students on the Master’s program in public health) were supervised in the field by assistants from the Kinshasa School of Public Health during data collection. The interviews were conducted in the language commonly spoken in Kinshasa, Lingala, or in French for those who wished to do so.

### 2.5. Statistical Analysis

The data were collected and sent to a dedicated server. Following a thorough examination of quality and consistency, the data were transferred to Stata 17 (StataCorp, College Station, TX, USA) for analysis. Descriptive statistics were employed to elucidate the fundamental characteristics of the study data. Means and standard deviations (SDs) were computed for continuous variables that followed a normal distribution, whereas proportions with their corresponding 95% confidence intervals (CIs) were determined for categorical variables. The connections between the dependent variables and each independent variable were tested using either Pearson's chi-square test or Fisher's exact test. Logistic regression was employed to ascertain the factors correlated with hesitation towards COVID-19 immunization. Variance inflation factors (VIF) were calculated to evaluate multicollinearity, and a VIF threshold of less than 4 was used to determine the absence of multicollinearity. All tests were conducted with a significance level of α = 0.05.

### 2.6. Ethical Approval

The protocol used in this study received ethical approval from the School of Health Ethics Committee (reference number: ESP/CE/71B/2022). Participants were informed that their participation was voluntary and that they could withdraw from the study without any consequences. It should be noted that oral informed consent was obtained from each participant. Participants were informed that participating in this survey would not guarantee any immediate benefit and that the results of this study could help the Expanded Programme on Immunisation and the Multisectoral Committee for the Control of COVID-19 to implement evidence-based interventions for the prevention and control of the COVID-19 pandemic. Confidentiality was ensured by maintaining the anonymity of the study participants.

## 3. Results

### 3.1. Characteristics of Participants

Table 1 displays the socio-demographic attributes of the individuals involved in the study. In total, 28.9% of the respondents belonged to the age ranges of 25–34 and 35–49, making them the largest demographic segments. Women constituted a significant portion of the participants, making up 69% of the sample. Among religious affiliations, 48.7% of the participants were associated with revivalist churches, followed by the Catholic church at 19.2%. A substantial proportion of the respondents had attained a secondary education level (43.1%). Notably, nearly half of the respondents (49.3%) were occupied as housewives, students, or pupils.

### 3.2. COVID-19 Vaccination Uptake

Overall, 15% (CI 95%: 13.25–17.9) of respondents had received at least one dose of a COVID-19 vaccine.

### 3.3. Hesitancy toward COVID-19 Vaccine

Table 2 presents data on the hesitancy to receive the COVID-19 vaccine based on the socio-demographic characteristics of the participants. The prevalence of vaccine hesitancy was 67% (CI95%:64.9–69.1). It indicates that younger respondents exhibited higher levels of vaccine hesitancy (74.4%) compared to older individuals. Additionally, women demonstrated greater vaccine hesitancy than men, and this difference was statistically significant (*p* < 0.001). Surprisingly, those who did not reside in slum areas showed the highest level of vaccine hesitancy.

### 3.4. Reasons for Refusal and Acceptance of COVID-19 Vaccine

Figure 1 illustrates the reasons that the participants gave for refusing vaccination. The reason given most often by the respondents was concerns about the vaccine’s safety and potential adverse effects, accounting for 45% of refusals. This was followed by doubts about the vaccine’s effectiveness (11%) and religious objections (9%).

Regarding the reasons for accepting the vaccine (Figure 2), 26% of respondents believed that the vaccine offered protection against reinfection, while 22% considered it safe. Additionally, approximately 7% felt compelled to accept the vaccine due to their close contact with vulnerable individuals whom they wished to protect.

As shown in Figure 3, respondents under the age of 25 were 2.15 times more likely to exhibit vaccine hesitancy (95% CI: 1.43–3.24; *p* < 0.001) compared to those aged 65 years and older. Similarly, respondents aged 25–34 were 1.86 times more likely to exhibit vaccine hesitancy (95% CI: 1.27–2.73; *p* = 0.001) compared to those aged 65 and older. Female respondents were 1.37 times more likely to exhibit vaccine hesitancy (95% CI: 1.11–1.70; *p* = 0.004) compared to male respondents. Non-slum households were 1.39 times more likely to exhibit vaccine hesitancy (95% CI: 1.09–1.75; *p* = 0.006) compared to slum households.

This community-based cross-sectional survey, carried out in Kinshasa, DR Congo, aimed to assess COVID-19 vaccine coverage and identify the reasons why hesitancy is shown toward these vaccines within the population. This study found that less than a quarter of respondents had received at least one dose of a COVID-19 vaccine. Among the reasons given for refusing to be vaccinated, most respondents cited concerns about the vaccine’s safety or adverse reactions. Among the reasons given for accepting the vaccine, a quarter of respondents believed that it prevented reinfection. The factors associated with vaccine hesitancy included female gender, an age of less than 35 years, and living in non-slum households.

This study’s findings regarding vaccine coverage were much lower than those of a study conducted by Ditekemena et al., which found that approximately 41% of the population in the city of Kinshasa were willing to be vaccinated if offered the vaccine [16]. This proportion is significantly lower than that observed in our study, where only a small number of people in Kinshasa had received at least one dose. However, it should be noted that a small proportion of the population was obligated to accept the vaccine due to the requirements of most European countries [22,23]. The socio-economic level of the majority of the population did not allow them to travel internationally and they did not want to be vaccinated.

The main reason given for vaccine hesitancy in our study was concerns about the vaccine’s safety or adverse reactions. Our results align with those of other studies, including that conducted by Okubo et al. in Japan; this study found that adverse reactions were the main reason for individuals exhibiting hesitancy toward vaccination [24]. A study conducted by Campelo in Brazil also found that the fear of adverse reactions was the primary obstacle to vaccine uptake [25]. We believe that this may not only be due to rumors about the adverse effects of certain vaccines, such as Astra Zeneca, but also due to the widespread media coverage of these rumors. In the DRC, for example, this situation led to an initial suspension of vaccination before it resumed. This information was sometimes disseminated by the scientific communities as well [20].

In Egypt, medical students stated that they were not vaccinated due to their fear of adverse effects, vaccine ineffectiveness, and a lack of information about different vaccines [25]. Moreover, the vaccine was accepted so as to avoid reinfection. Belief in the existence of the disease is a prerequisite for the acceptance of all preventive measures. Therefore, it is important to reinforce communication strategies in order to help people understand the benefits of vaccination. Farooq Ahmad et al. found that vaccination was accepted in their study due to the belief that it would stop the pandemic [25]. This highlights the importance of effective communication strategies. Effective strategies should also be implemented in our context, as the majority of the population did not believe in the existence of the disease, making it difficult for them to adhere to various interventions.

In our study, females, people under the age of 35, and those living in non-slum households were more likely to exhibit vaccine hesitancy. The female gender and an age of less than 35 have been found to be associated with hesitancy in several studies [24,26]. A study conducted by Soares in Portugal found that young people were more hesitant to be vaccinated compared to older people [26]. In China, Xiao et al. found that young adults aged between 18 and 34 were the most reluctant to be vaccinated [27]. We believe that this may be because the impacts that were experienced by the younger age group during the pandemic were less severe in most countries, even though they were able to transmit the disease to at-risk people, including the elderly or people with comorbidities such as diabetes, hypertension, and other chronic diseases. This younger population often did not perceive themselves as being at risk. Additionally, at the start of the COVID-19 vaccination program in the DRC, certain priority groups were targeted first, including the elderly, healthcare professionals, and people with comorbidities [20]. As a result, it was difficult to vaccinate enough young people.

Females were more likely to exhibit reluctance with regard to vaccination than males. We think that this could be explained by the fact that women are more likely to believe rumors and are more fearful than men. Rumors about vaccination appeared to have a greater impact on women, who were more afraid of the potential adverse effects of the vaccine [28]. A similar reluctance regarding vaccination among women has been observed in other developing countries like Senegal and Ethiopia [29,30].

Rumors generally circulate among those who are the most informed and those who use social networks [31]. These channels have been a vehicle for many rumors about the COVID-19 vaccine [32]. This could explain why individuals from non-slum households, who have a higher socio-economic status and more access to these media, are more hesitant to be vaccinated. On the other hand, people living in non-slum areas may be more likely to accept the information conveyed by community health workers and adhere to vaccination.

These findings are constrained by many limitations. While the exact reasons for the higher estimations found from our survey compared to the National Immunization Program (NIP) data cannot be determined, there are various potential causes. Regarding the survey data, it is important to note that the COVID-19 vaccination status was determined based on self-reported information, which may not have accurately reflected the actual vaccination status of some respondents. Second, the residents of Kinshasa Province who received their vaccinations elsewhere may have been excluded from the Kinshasa vaccine administration data (NIP). This could account for some of the significant disparities reported between the immunization data and survey data. Third, it is possible that there were COVID-19 vaccinations taking place during the current campaign (from 20 July 2022) that have not been reported in the administration data (NIP). This would artificially decrease the coverage estimates derived from the vaccine administration data (NIP); however, this could potentially represent a minuscule numerical value. Furthermore, the NIP does not contain accurate data about the population size (the last census occurred in 1984). Fifth, these findings are relevant to only the data from July, and any potential bias may have since been altered.

Finally, social desirability bias could lead to some unvaccinated individuals claiming that they are vaccinated. It is, thus, likely that the population survey overestimates vaccination coverage due to survey respondents misreporting their vaccination status. The hesitancy discussed in this research may simply reflect the subjective viewpoints of individuals, rather than being a true representation of hesitancy. This is because we lack evidence to confirm if the vaccine was readily accessible or if healthcare personnel actively contacted individuals with vaccination kits and a planned schedule. The study's findings are specific to Kinshasa and may not be generalizable to other provinces. However, this study gives a clear picture of COVID-19 vaccination coverage while the vaccine was available and while all awareness-raising strategies were in place. This study will thus enlighten the National Public Health Institute and the NIP in the DRC in the process of vaccine acceleration as envisaged in the document for integrating the complete package of activities to combat COVID-19 into primary healthcare. Knowledge of the reasons for vaccine hesitancy will help reduce the risk of large quantities of vaccine running out and boost vaccination of targets in provinces with low vaccine coverage. The results will enable the Ministry and all stakeholders to improve the strategies that are implemented and thus increase vaccination coverage based on the factors associated with the population’s hesitancy regarding vaccination.

## 4. Conclusions

The COVID-19 pandemic had a significant impact on people's way of life, posing a humanitarian challenge. Vaccination has been instrumental in reducing the spread of this worldwide health emergency. The current COVID-19 vaccine coverage in Kinshasa is insufficient and there is a significant frequency of vaccine reluctance. Despite the implementation of many nationwide measures, vaccination hesitancy continues to be a significant concern. To enhance vaccine acceptance and boost immunization rates within the targeted population, stakeholders must consider the reasons that contribute to both vaccination refusal and vaccine acceptance. Furthermore, it is imperative to bolster awareness-raising initiatives throughout the community in order to increase the percentage of acceptance of vaccination. Future studies should address the population’s perceptions of the COVID-19 vaccine. Similar studies would be important at the national level and should include all provinces or, better yet, all health zones to better describe this phenomenon at the national level.

## Figures and Tables

**Figure 1 vaccines-12-00188-f001:**
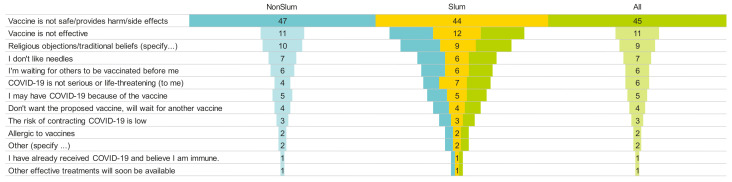
Reasons for refusing the COVID-19 vaccine (in %).

**Figure 2 vaccines-12-00188-f002:**
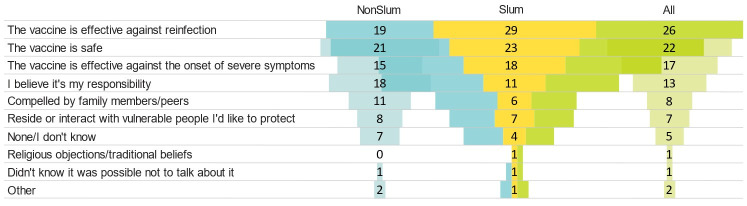
Reasons for accepting the COVID-19 vaccine: 3.5. Factors Associated to the Hesitancy of COVID-19 Vaccine (in %).

**Figure 3 vaccines-12-00188-f003:**
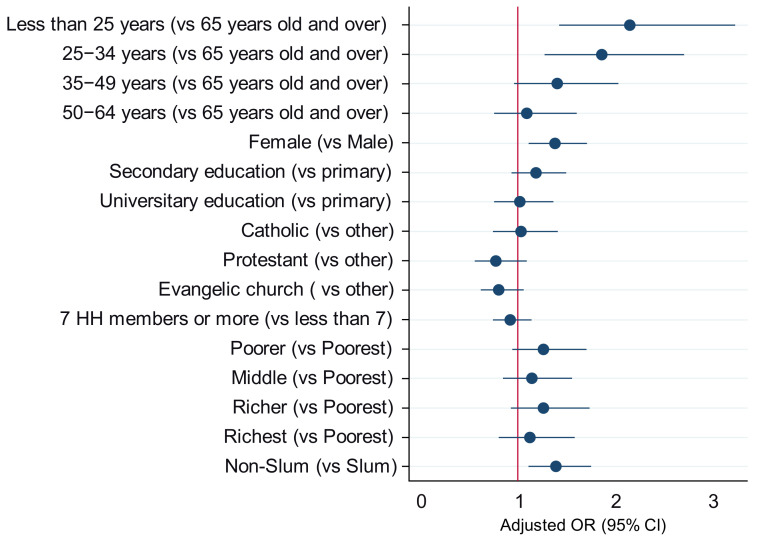
Forest plot: factors associated with hesitancy toward the COVID-19 vaccine and discussion.

**Table 1 vaccines-12-00188-t001:** Characteristics of the participants.

	No Slum		Slum		Ensemble		*p*
	*n*	%	*n*	%	*n*	%	
Age							0.299
<25	94	14.7	238	17.6	332	16.6	
25–34	189	29.5	387	28.6	576	28.9	
35–49	180	28.1	396	29.2	576	28.9	
50–64	112	17.5	221	16.3	333	16.7	
≥65	66	10.3	112	8.3	178	8.9	
Gender							0.492
Male	196	30.0	430	31.5	626	31.0	
Female	458	70.0	936	68.5	1394	69.0	
Level of Education							<0.001
None/Primary School	110	16.9	592	43.4	702	34.8	
Secondary School	299	45.9	570	41.8	869	43.1	
University/High	243	37.3	202	14.8	445	22.1	
Religion of respondent							<0.001
Catholic	160	24.5	228	16.7	388	19.2	
Protestant	111	17.0	172	12.6	283	14.0	
Revival Church	284	43.4	699	51.2	983	48.7	
Others	99	15.1	267	19.5	366	18.1	
Employment							<0.001
No occupation/housewife/student or pupil	313	47.9	683	50.0	996	49.3	
Public sector employee with a regular monthly salary	84	12.8	122	8.9	206	10.2	
Private sector employee with a regular monthly salary	41	6.3	65	4.8	106	5.2	
Self-employed in the private sector (self-employed)	119	18.2	170	12.4	289	14.3	
Worker in the informal sector and small trade	71	10.9	262	19.2	333	16.5	
Agropastoral and fishing	3	0.5	12	0.9	15	0.7	
Other	23	3.5	52	3.8	75	3.7	
Household size							<0.001
≤6	548	83.8	966	70.7	1514	75.0	
≥7	106	16.2	400	29.3	506	25.0	
Sufficient living space (not overcrowded)							<0.001
Overcrowding	875	29.1	3167	42.9	4042	39.0	
Sufficient Living Area	2127	70.9	4208	57.1	6335	61.0	
Income Quintiles							<0.001
Very low	44	6.7	360	26.4	404	20.0	
Low	74	11.3	330	24.2	404	20.0	
Middle	124	19.0	281	20.6	405	20.0	
High	159	24.3	244	17.9	403	20.0	
Very High	253	38.7	151	11.1	404	20.0	

**Table 2 vaccines-12-00188-t002:** Hesitancy to receive the COVID-19 vaccine.

	No		Yes		*p*
	*n*	%	*n*	%	
Age					<0.001
<25	85	25.6	247	74.4	
25–34	161	28.0	415	72.0	
35–49	199	34.5	377	65.5	
50–64	137	41.1	196	58.9	
≥65	76	42.7	102	57.3	
Gender					<0.001
Male	242	38.7	384	61.3	
Female	424	30.4	970	69.6	
Level of Education					0.023
None/Primary School	256	36.5	446	63.5	
Secondary School	260	29.9	609	70.1	
University/High	148	33.3	297	66.7	
Religion of respondent					0.674
Catholic	128	33.0	260	67.0	
Protestant	97	34.3	186	65.7	
Revival Church	330	33.6	653	66.4	
Others	111	30.3	255	69.7	
Employment					0.262
No occupation/housewife/student or pupil	306	30.7	690	69.3	
Public sector employee with a regular monthly salary	75	36.4	131	63.6	
Private sector employee with a regular monthly salary	39	36.8	67	63.2	
Self-employed in the private sector (self-employed)	99	34.3	190	65.7	
Worker in the informal sector and small trade	121	36.3	212	63.7	
Agropastoral and fishing	6	40.0	9	60.0	
Other	20	26.7	55	73.3	
Household size					0.184
≤6	487	32.2	1027	67.8	
≥7	179	35.4	327	64.6	
Sufficient living space (not overcrowded)					0.770
Overcrowding	197	33.4	392	66.6	
Sufficient Living Area	469	32.8	962	67.2	
Income Quintiles					0.158
Very low	154	38.1	250	61.9	
Low	128	31.7	276	68.3	
Middle	132	32.6	273	67.4	
High	122	30.3	281	69.7	
Very High	130	32.2	274	67.8	
Slum Household					0.001
No	183	28.0	471	72.0	
Yes	483	35.4	883	64.6	

## Data Availability

The dataset used for analysis can be obtained upon reasonable request by writing an email to the corresponding author.

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
