# Peer review of "COVID-19 Vaccine Coverage and Factors Associated with Vaccine Hesitancy: A Cross-Sectional Survey in the City of Kinshasa, Democratic Republic of Congo"

_vaccines, 2024, doi:10.3390/vaccines12020188_

Round 1

Reviewer 1 Report

Comments and Suggestions for Authors

The authors provided a comprehensive analysis of COVID-19 vaccine uptake and hesitancy in Kinshasa. Utilizing a detailed cross-sectional survey approach and employing Lot Quality Assurance Sampling (LQAS) for data collection, the study aimed to quantify vaccine coverage and identify key factors that contribute to vaccine hesitancy. This investigation is crucial for understanding the barriers to vaccine acceptance and designing targeted public health interventions in the region.

1. The study's findings are specific to Kinshasa and may not be generalizable to other regions. A clearer discussion on the limitations regarding generalizability would be useful.

2. The study's reliance on self-reported vaccination status raises concerns about the accuracy of this data. Exploring methods to validate these self-reports or acknowledging this limitation more explicitly would strengthen the study.

Author Response

Authors ‘Responses

The authors provided a comprehensive analysis of COVID-19 vaccine uptake and hesitancy in Kinshasa. Utilizing a detailed cross-sectional survey approach and employing Lot Quality Assurance Sampling (LQAS) for data collection, the study aimed to quantify vaccine coverage and identify key factors that contribute to vaccine hesitancy. This investigation is crucial for understanding the barriers to vaccine acceptance and designing targeted public health interventions in the region.

Response: thanks

  1. The study's findings are specific to Kinshasa and may not be generalizable to other regions. A clearer discussion on the limitations regarding generalizability would be useful.

Response: thanks, this is now well explicitly stated in the current version.

  1. The study's reliance on self-reported vaccination status raises concerns about the accuracy of this data. Exploring methods to validate these self-reports or acknowledging this limitation more explicitly would strengthen the study.

Response: Thanks, this was discussed as one of the limitations.

See “First, regarding the survey data, the COVID-19 vaccination status was based on self-reports, which may not have accurately reflected the actual vaccination status of some respondents.”

Reviewer 2 Report

Comments and Suggestions for Authors

Akilimali et al. have studied COVID-19 vaccination coverage and vaccine hesitancy in Kinshasa (Congo).

I found the following strange thing in the article. 1. In Abstract part, why background title appearing immediately. Same is with method, results, and conclusions. I checked articles published in “vaccines” (MDPI) journal, such pattern does not exists.

1. In Africa, to be fair, the vaccine was not available as was the case in the US, Europe, Japan, Israel, and other countries that made vaccine available to all of their citizens. So, how the authors have justified if it was a case of hesitancy or unavailability of vaccine? Show data that even though vaccine was available, there was hesitancy for vaccination. The data shown in the paper has not proved if vaccine was at hand (available) or health staff approached with vaccination kits and planned schedule. It is just a survey to know the opinion of the people, not the real hesitancy.

2. As mentioned in Introduction about source, the Covax supply (WHO) was very limited, not enough.

3. Specify DRC before using a short form. Smaller paragraphs in Introduction must be combined.

The paper needs a major revision to address the above issues

Comments on the Quality of English Language

Needs improvement. 

Author Response

Authors ‘Responses

Comments and Suggestions for Authors

Akilimali et al. have studied COVID-19 vaccination coverage and vaccine hesitancy in Kinshasa (Congo).

I found the following strange thing in the article. 1. In Abstract part, why background title appearing immediately. Same is with method, results, and conclusions. I checked articles published in “vaccines” (MDPI) journal; such pattern does not exist.

Response: Thanks, we have removed in the current version

  1. In Africa, to be fair, the vaccine was not available as was the case in the US, Europe, Japan, Israel, and other countries that made vaccine available to all of their citizens. So, how the authors have justified if it was a case of hesitancy or unavailability of vaccine? Show data that even though vaccine was available, there was hesitancy for vaccination. The data shown in the paper has not proved if vaccine was at hand (available) or health staff approached with vaccination kits and planned schedule. It is just a survey to know the opinion of the people, not the real hesitancy.

Response: Thanks, we agree with the reviewer. However, the way we defined hesitancy included also availability aspect. Vaccine hesitancy was defined as “delay in acceptance or refusal of vaccination despite the availability of vaccination services”. In this study, vaccine hesitancy was defined as the response of “no” or “don’t know/not sure” to whether the participant would get the COVID-19 vaccine as soon as it became available.

We then asked respondent the reasons of refusal of COVID-19 Vaccine, availability of vaccine was not cited by the respondent. (see figure Figure 1. Reasons for refusing the COVID-19 vaccine (in %).)

  1. As mentioned in Introduction about source, the Covax supply (WHO) was very limited, not enough.

Response: Yes, we have stated this.

  1. Specify DRC before using a short form. Smaller paragraphs in Introduction must be combined.

Response: Thanks, and we have edited this, accordingly

Comments on the Quality of English Language: Needs improvement. 

This manuscript was sent to MDPI and has undergone English language editing by MDPI. The text has been checked for correct use of grammar and common technical terms and edited to a level suitable for reporting research in a scholarly journal. MDPI uses experienced, native English-speaking editors. The certificate number is:  english-76440.

Round 2

Reviewer 2 Report

Comments and Suggestions for Authors

As I wrote in my first report, this paper is not on vaccine hesitancy. The title must be changed. This is just a survey of the individuals on what they think about vaccination. Authors must prove there were sufficient doses available to vaccinate the population. Since the WHO and other organizations could provide only limited donated shots never sufficient for the entire African or Congo population. The survey staff did not approach the population with vaccines at hand. There is no such mention or data in the paper. Are the doses/stock still available at present for the entire population?  Explain in data, figures and numbers.   

Comments on the Quality of English Language

First, provide the dose availability data, English needs to be improved. 

Author Response

Comments and Suggestions for Authors

As I wrote in my first report, this paper is not on vaccine hesitancy. The title must be changed. This is just a survey of the individuals on what they think about vaccination. Authors must prove there were sufficient doses available to vaccinate the population. Since the WHO and other organizations could provide only limited donated shots never sufficient for the entire African or Congo population. The survey staff did not approach the population with vaccines at hand. There is no such mention or data in the paper. Are the doses/stock still available at present for the entire population?  Explain in data, figures and numbers.   

Comments on the Quality of English Language

First, provide the dose availability data, English needs to be improved.

We have provided dose availability data which is stated as follow:

“According to the National Immunization Program's data as of July 31, 2022, out of the 39,929,390 doses received in the country, 1,445,060 AstraZeneca doses were returned to COVAX. Additionally, 143,300 doses expired at the Central Hub, with 130,000 doses being Turkovac and 13,300 doses being AstraZeneca. Furthermore, 36,825,190 doses were distributed to the provinces. The Democratic Republic of Congo (DRC) possessed a total of 6,529,628 doses of a certain substance. This included 1,515,840 doses located in the Central Hub, and 5,013,788 doses distributed among the provinces. A total of 2,182,951 doses throughout the provinces had reached their expiration date.  The government has administered a total of 19,753,514 doses of the vaccine, as of July 31, with a reported completion rate of 76.3%. “ 

We than added this statement in the discussion section as one of the limit of this study: “The hesitancy discussed in this research may simply reflect the subjective viewpoints of individuals, rather than being a true representation of hesitancy. This is because we lack evidence to confirm if the vaccine was readily accessible or if healthcare personnel actively contacted individuals with vaccination kits and a planned schedule.”

Regarding English: We will request again MDPI service to edit the manuscript.

Round 3

Reviewer 2 Report

Comments and Suggestions for Authors

1. As, I mentioned before, short paragraphs should be merged into large paragraphs in a research paper. Improve the Introduction part.  There are no strong reasons for changing the paragraph every few sentences. Two parts of short paragraphs can be combined. 

2. Make conclusions large. The conclusion is shorter than the abstract, strange! 

3. Vaccine availability data has been included, now OK. 

Comments on the Quality of English Language

Same as in the authors' section. 

Author Response

  1. As, I mentioned before, short paragraphs should be merged into large paragraphs in a research paper. Improve the Introduction part.  There are no strong reasons for changing the paragraph every few sentences. Two parts of short paragraphs can be combined. 

Response: We have combined short paragraphs

  1. Make conclusions large. The conclusion is shorter than the abstract, strange! 

Response: We have edited the conclusion which is now stated as follow:

The COVID-19 pandemic had a significant impact on people's way of life, posing a humanitarian challenge. Vaccination has been instrumental in reducing the spread of this worldwide health emergency. The current COVID-19 vaccine coverage in Kinshasa is insufficient, and there is a significant frequency of vaccine reluctance. Despite the implementation of many nationwide measures, vaccination hesitancy continues to be a significant concern. To enhance vaccine acceptance and boost immunization rates within the targeted population, stakeholders must consider the reasons that contribute to both vaccination refusal and vaccine acceptance. Furthermore, it is imperative to bolster awareness-raising initiatives throughout the community in order to increase the percentage of acceptance of vaccination. Future studies should address the population’s perceptions of the COVID-19 vaccine. Similar studies would be important at the national level, and should include all provinces or, better yet, all health zones, to better describe this phenomenon at the national level.

  1. Vaccine availability data has been included, now OK. 

Response:  Thanks

Comments on the Quality of English Language: Same as in the authors' section.

Response: We are requesting English language editing by MDPI as we did before submitting